# In Silico Analysis of Gene Expression Change Associated with Copy Number of Enhancers in Pancreatic Adenocarcinoma

**DOI:** 10.3390/ijms20143582

**Published:** 2019-07-22

**Authors:** Rajesh Kumar, Sumeet Patiyal, Vinod Kumar, Gandharva Nagpal, Gajendra P.S. Raghava

**Affiliations:** 1Department of Computational Biology, Indraprastha Institute of Information Technology, Delhi 110020, India; 2Bioinformatics Centre, CSIR-Institute of Microbial Technology, Chandigarh 160036, India

**Keywords:** enhancer, copy number variation, regulatory elements, pancreatic cancer, adenocarcinoma, super enhancer, survival, differential expressed genes, functional enrichment analysis

## Abstract

Understanding the gene regulatory network governing cancer initiation and progression is necessary, although it remains largely unexplored. Enhancer elements represent the center of this regulatory circuit. The study aims to identify the gene expression change driven by copy number variation in enhancer elements of pancreatic adenocarcinoma (PAAD). The pancreatic tissue specific enhancer and target gene data were taken from EnhancerAtlas. The gene expression and copy number data were taken from The Cancer Genome Atlas (TCGA). Differentially expressed genes (DEGs) and copy number variations (CNVs) were identified between matched tumor-normal samples of PAAD. Significant CNVs were matched onto enhancer coordinates by using genomic intersection functionality from BEDTools. By combining the gene expression and CNV data, we identified 169 genes whose expression shows a positive correlation with the CNV of enhancers. We further identified 16 genes which are regulated by a super enhancer and 15 genes which have high prognostic potential (*Z*-score > 1.96). Cox proportional hazard analysis of these genes indicates that these are better predictors of survival. Taken together, our integrative analytical approach identifies enhancer CNV-driven gene expression change in PAAD, which could lead to better understanding of PAAD pathogenesis and to the design of enhancer-based cancer treatment strategies.

## 1. Introduction

Cancer is a genetic disease, as initiation, progression, and metastasis are governed by several genetic and epigenetic changes within the genome. Pancreatic cancer is a leading cause of mortality in the Western world [1]. Patients are mostly diagnosed at an advanced stage, resulting in poor response to therapy. The 5-year survival rate of pancreatic cancer patients is 6%. This is the worst survival rate among all 22 common types of cancer [2]. Cancer cells harbor thousands of genomic alterations, including amplification, deletion, insertion, translocation, transversion and copy number variation (CNV). Only a small fraction of these genomic alterations represents the driver mutations that are truly morbific. Often biological function of a cell is maintained by a complex multi-level gene regulatory hierarchy, including post-translation modification, enhancer activation, miRNA mediated gene regulation, and RNA editing. In cancer cells, these regulatory circuits are disrupted or rewired, which leads to the disease phenotype [3].

Whole genome sequencing of tumor and normal samples provides unprecedented data that can be used to identify mutations associated with disease. The Cancer Genome Atlas (TCGA) [4] and International Cancer Genome Consortium (ICGC) are such large-scale efforts [5]. These projects and follow up studies identified several mutated genes and pathways, and have significantly increased our knowledge of cancer, leading to the discovery of new targets, diagnoses, prognoses and improvements of therapy. Mutation in the coding region of several genes has been consistently identified by several studies, such as with altered frequency (e.g., *KRAS, TP53*) or loss (e.g., *CDNK2A*, *SMAD4*, *ROBO2*), and altered pathways, such as Wnt/Notch. Thus far, cancer studies have mostly focused on characterizing the functional impact of mutations in protein-coding sequences. COSMIC (Catalogue of Somatic Mutation in Cancer) is one such effort, which lists only aberrations in the coding sequence of the genes [6].

The coding fraction represents only 2% of the genome. The remaining 98% the of human genome is transcribed either in non-coding RNA or into regulatory elements. Most of the genomic alteration resides in the non-coding region of the genome [7]. It is also observed that more than 80% of genetic variants associated with diseases were observed in the non-coding region of the genome [8]. Nevertheless, this portion of the genome has been largely unexplored. Mutation in the non-coding region of the genome can modify the function of both cis- and trans-acting elements in a regulatory circuit [9,10]. Therefore, this can lead to the development of cell behavior towards tumorigenesis [11]. However, our knowledge of gene regulatory circuit rewiring is far from complete. Existing research shows the importance of the enhancers as a key piece in this gene regulation circuit [12].

Enhancers are DNA elements of up to 50–1500 base pairs (bp) [13]. They interact with their target promoters irrespective of their position to regulate downstream gene expression [14]. Several studies link alteration within the enhancer to disease phenotype, e.g., DiseaseEnhancer is one such database containing information on 847 disease associated enhancers in 143 human diseases [15]. Several studies confirm the dosage effect of copy number variation in gene expression [16,17]. The dosage effect of loss/gain in enhancer elements is further supported by experimental evidence. Zhang X., et al. [18] demonstrate that duplication in the enhancer region results in higher gene expression by luciferase assay. Experimental studies also confirm that copy number alterations within the enhancer element are an important driver of tumorigenesis. In head and neck squamous cell carcinoma, KLF5 was upregulated by amplification super-enhancers marked by H3K27ac [19].

In this study, we represent an integrative approach that embellishes the relationship between enhancer CNV and targeting gene expression changes in pancreatic adenocarcinoma. The following are main goals of this study: (i) identification of enhancers having a significant change in their CNVs; (ii) understanding the effect of a change in enhancer CNVs on the expression of associated genes; (iii) understanding the clinical potential of genes regulated by CNVs of enhancers; and, (iv) pathway enrichment analysis of enhancer associated genes. In order to achieve our objectives, we performed the following steps. Firstly, coordinates of enhancers specific to pancreatic tissue, as well as the associated genes of these enhancers, were obtained from EnhancerAtlas [20]. EnhancerAtlas provides high quality experimental data on enhancers and cross validation for each cell/tissue type was done by integration of the multiple experimental dataset [21]. Secondly, gene expression and CNV data of 185 patients were obtained from TCGA. Thirdly, the software GISTIC was used for identifying CNVs having a significant change (loss/gain) in PAAD samples. Fourthly, we used BEDTools to identify enhancers specific to pancreatic tissue having significant variation in their CNVs in PAAD samples. These enhancers were further classified based on their increase or decrease in CNVs and based on positive or negative correlation with associated genes. Finally, we performed a wide range of analyses to understand the effect of enhancers on gene regulation, with the potential to classify high and low risk patients, and the rewiring of regulation in cancer.

## 2. Results

### 2.1. Identification of Enhancers Having Change in CNV

In total, 952 significant CNV regions were obtained for the pancreatic cancer sample, out of which 321 regions were amplified and 631 regions were deleted (Appendix A). Based on the genomic coordinate, by employing genomic intersecting functionality from BEDTools, we mapped enhancer coordinates on to the significant CNV region. Finally, 421 enhancers having a significant variation in their CNV region (amplified/increase or deleted/decreased) were mapped on to enhancer coordinates (Appendix A). The gene expression analysis of quantile normalized pancreatic cancer vs normal samples identified 2431 upregulated and 3614 downregulated genes. By checking the expression of enhancer target genes in the same TCGA sample, we identified 169 concordant regulatory pairs, which reflected a CNV-based enhancer dosage effect on gene expression (Appendix A). This regulatory pair consisted of 89 upregulated and 80 downregulated genes; all 169 gene expressions showed a positive correlation with the CNV of enhancers. Our analysis in this study mainly focused on these 169 genes, and considered two possible concordant changes: enhancer copy number gain/gene upregulation and copy number loss/gene downregulation.

### 2.2. Pathway Enrichment Analysis of Genes Regulated by Enhancers

In order to gain further insight into the enhancer CNV-driven differentially expressed genes, gene interaction network and enrichment analyses were performed using STRING (Search Tool for the Retrieval of Interacting Genes/Proteins) and Enrichr, respectively. As shown in Figure 1, upregulated genes RPS16, SCAMP3 and MUC1 show three or more interaction direct interactions. These genes are the major components regulating the glucose metabolism and proteolysis in pancreatic cells [22,23], indicative of poor prognosis [24] and involved in *PDGFA* expression in pancreatic cancer progression, respectively [25]. Figure 2 (for downregulated) shows *UBB*, *PRPF8*, *ERLIN* and *RPS6* genes involved in direct interaction. The *UBB* gene is involved in TNF-α induced NF-κB activation, and thus plays a role in the stabilization of the tumor suppressor *P53* gene. Thus, downregulation of the UBB gene by any means may serve as an anti-tumor treatment [26]. RPS6 and *ERLIN1* genes are involved in the mTOR signaling pathway, which is essential for cell growth and metabolism, particularly tumor formation and angiogenesis [27,28]. Pathway enrichment analysis of the concordant gene set provides clues regarding their role in regulating cell cycle progression, and the Eph cell signaling pathway (Figure 3). This pathway regulates kinase activity, which is frequently mutated in almost all types of cancer [29]. GO (Gene Ontology) enrichment analysis was performed to identify the most correlative biological, molecular and cellular function of the genes. GO analysis demonstrates that the genes are enhanced in receptor binding and in regulation of the apoptotic pathway. The detailed results are presented in Figure 4.

### 2.3. Cox Proportional Hazard Analysis of Enhancer Regulated Genes

Univariate analysis of genes was done using a Cox proportional hazard regression model on the mean and median cut off of gene expression across the patient samples. On performing the analysis, 41 out of 169 genes were found to be significant on the basis of *p*-value < 0.05, which is statistically significant. Twenty-nine genes had a hazard ratio of more than 1.5, that is, the expression of these genes results in the progression of cancer at the rate of more than 1.5 in patients belonging to a high-risk group [expression value < median(gene expression)] as compared to the low-risk group [expression value > median(gene expression)]. Table 1 Shows the top 15 genes. *VAMP2* has a HR of 2.9, and downregulation of genes is involved in acute acinar pancreatitis [30]. The *CERK* gene, with a HR of 2.48, is involved in metastasizing of pancreatic cancer cells and thus leads to cancer development at distant places [31].

### 2.4. Prognostic Potential of Regulatory Genes

The prognostic potential of genes regulated by enhancer CNV was identified using data from the human prognostic database PRECOG [32]. Genes with a *Z*-score >1.96 can be evaluated as a prognostic biomarker as this is equivalent to a two tailed *p*-value < 0.05. We identified 15 genes of the 169 concordant regulatory pair genes having a z-score greater than 1.96, as shown in Table 2 and Appendix A. Among the prognostic genes, several genes are involved in a number of pathways regulating pancreatic cancer development: the gene UCK2 is vital for regulating apoptosis; the *HDGF* gene is considered a jack of all trades in cancer [33]; *SSR2* gene inhibition lowers cancer loads in vitro [34]; the *USP21* gene is the master regulator of the Hippo pathway [35]; MUC16 regulates pancreatic cancer cell metastasis [36]; and, the *STOML2* gene is a prognostic biomarker for pancreatic cancer [37].

### 2.5. Genes Regulated by Multiple Enhancers or Super Enhancers

It has been observed that multiple enhancers or clusters of enhancers regulate a single gene. A gene regulated by multiple enhancers is called a gene regulated by super enhancers. We identified 16 genes which are regulated by super enhancers; the Ensembl ID of genes and genomic coordinates of its enhancers are shown in Table 3. There are two main types of aberrant super-enhancers found in various cancers: those involving mutations generated in super-enhancers and those involving the acquisition of new oncogenic super-enhancers. In our study we have identified super-enhancer regions that are created due to the amplification event. These are the master regulators of the cell’s fate and identity. Inhibition of super-enhancers seems to be an effective therapy for lowering the burden of cancer, and overexpression of genes regulated by these can act as a diagnostic marker for cancer progression [38].

### 2.6. Enhancer Expression Correlation with Patient Survival

To unravel the complex relationship between expression and survival, we analyzed the enhancer expression correlation with patient survival, to display the positive or negative correlation between the two. The enhancer locations, which show the maximum divergence between cancer and the normal sample, are listed in Table 4 and Appendix A.

### 2.7. Prognostic Potential of Negatively Correlated Genes

In this study, our major emphasis was on 169 positively correlated genes (data shown in Table 1), thus we used these genes for all analysis, including identification of prognostic biomarkers. The aim was to understand the impact of negatively correlated genes, meaning enhancer copy number loss/gene upregulation or enhancer copy number gain/gene downregulation (E^+^G^−^/E^−^G^+^). These genes are negatively correlated with CNV of the enhancer. In order to understand the prognostic potential of these genes, we computed the Cox proportion hazard ratio with the *p*-value. The performance of the top genes is shown in Table 5 and performance of the remaining genes is shown in Appendix A.

## 3. Conclusions and Discussion

Gene expression and cell phenotype are governed by a complex set of regulatory circuits and enhancer elements, which are at the center of governing the fate of each cell cycle. For the better understanding of the disease phenotype and to provide better therapeutics, detailed knowledge of the mechanism of regulation is necessary for every gene whose misexpression is known to cause disease or is involved in disease progression. Due to the non-coding nature of the enhancers, their effect could only be seen through co-expressed genes. Our data integration approach of human enhancers and target provides the first comprehensive DNA–protein interaction in pancreatic cancer. By overlapping the information with CNV, we revealed several important aspects of enhancer deregulation in cancer development. The integration of data from matched TCGA allowed us to propose that the potential mechanism concerning the changes in gene expression is mediated by enhancer copy number gain or loss. We also identified genes (*HDGF*, *UBB*) which are upregulated, and downregulation of such genes lowers the cancer load. One such gene is *UBB*, and previous research shows downregulation of the *UBB* gene via siRNA and its anti-tumor effect in various cell lines and a mouse xenograft model. Thus we recommend downregulation of the *UBB* gene with enhancer mutation may deliver an anti-tumor effect for pancreatic cancer [26]. A number of super-enhancer linked target genes have been identified. The amplification of enhancers may be used for biomarkers in cancer and may be a potential target for anti-cancer drug design. One of the most fundamental approaches for treating genetic or epigenetic diseases is to disrupt or correct aberrant genomic sequences responsible for the generation of disease-associated enhancers. We also analyze the two gene sets: one shows a positive correlation with enhancers and the other shows a negative correlation with enhancers. From this comparative analysis, we concluded that although enhancers can bring about gene expression change, the survival of a patient with cancer is a multifaceted phenomenon, as the hazard ratio in genes that corresponds to enhancer nature is good but it is also good in genes which do not correspond to enhancer nature. It may be in the particular case of pancreatic cancer that genomic aberration may occur more in the genomic region than the regulatory region. Further study is required to confirm the results of survival of patients and enhancer expression. With recent advances in genome engineering technologies such as TALEN (Transcription activator-like effector nuclease) and CRISPR/Cas9, it is now more convenient to generate mutations in cells or animal models, providing unprecedented opportunities to develop effective gene therapies for enhancer-associated diseases. Moreover, enhancers and super-enhancers can be used as prognostic markers for the prediction of disease risk and progression. Thus, integrative analysis of a gene transcription signature and the enhancer profile of patients or healthy individuals could emerge as an important approach for disease diagnosis. We conclude that coupling of enhancer profiles with gene expression changes has possibly unearthed a powerful approach to treat disease, and can be expected to strengthen personalized medicine in the near future.

## 4. Materials and Methods

### 4.1. Enhancers and Target Gene for Human Pancreatic Tissue

In the present study, pancreatic tissue specific enhancers were mined from EnhancerAtlas (http://www.enhanceratlas.org/data/AllEPs/Pancreas_EP.txt). EnhancerAtlas maintains enhancers and associated/target genes of 105 human cell/tissue types. Enhancer sequences with their genomic coordinates mapped to human genome assembly GRCh37/hg19 were downloaded in plain text format. This contained 3876 sequences with genomic coordinates specific to pancreatic tissue. We also downloaded the target genes corresponding to these enhancers from EnhancerAtlas. Enhancers that do not interact with any promoter and gene sequence and with confidence scores below 0.7 were removed. This step resulted in only 1696 enhancers, which were strongly associated with their target gene and thus regulation of gene expression. These 1696 strong enhancers form 2968 interaction pairs. Figure 5 shows the overall mind map and computational workflow used to carry out this study.

### 4.2. Pancreatic Cancer Dataset from TCGA

The pancreatic adenocarcinoma (PAAD) dataset of copy number variations and gene expression levels of 185 samples was downloaded from The Cancer Genome Atlas (TCGA) database (https://portal.gdc.cancer.gov/projects/TCGA-PAAD). Out of 185 samples, 181 correspond to tumor samples and four samples consist of normal samples derived from adjacent solid tissue. More precisely, Illumina HiSeq RNASeqV2 in 728 files and copy number variations in 737 files were taken into consideration.

### 4.3. Copy Number Variations Identification

We implemented the GISTIC2.0 [39] (Genomic Identification of Significant Targets in Cancer, version 2.0) algorithm to identify the regions of the genome that were amplified or deleted across the samples. This algorithm uses human genome assembly Hg19 as the reference genome, which consists of information on recurrent CNV cytoband and gene location. Thresholds were set as default GISTIC2.0 parameters. The threshold for copy number gain/loss was set at 0.1, so that regions with a copy number value above 0.1 were acknowledged as a copy number gain (i.e., amplification), and regions with a copy number value below 0.1 were acknowledged as a copy number loss (i.e., deletion) [40]. Segments that comprised less than four markers were combined with the neighboring segment nearest in copy number; regions with q-values less than 0.25 were acknowledged as significant. We then used genomic intersecting functionality from BEDTools [41] to intersect each recurrent significant CNV identified from GISTIC to map onto the genomic coordinates of enhancers of pancreatic tissue. This step led to the identification of recurrent significant CNV of enhancers of pancreatic tissue.

### 4.4. Screening for Differential Gene Expression

Differential gene expression (DGE) analysis on the mRNA transcripts was done after quantile filtration, which was achieved using the quantile method and a cut-off value of 0.25. To ascertain whether a gene was expressed differentially, the test of the hypothesis, with the fold-change between the two conditions as normal and tumor conditions, was calculated. We applied the TCGAanalyze_DEA function, which performs DEA using various functions of the edgeR package from Bioconductor [42]. The function edgeR:exactTest makes pair-wise tests for differential expression between two groups. The *p*-values obtained from the exactTest sorted in ascending order were corrected/adjusted using the false discovery rate (FDR) correction which returns the top differentially expressed genes. Thresholds for log fold change (logFC) and FDR were set at 1 and 0.1, respectively, such that differentially expressed mRNAs were considered to be significant if logFC > 1 and FDR < 0.05.

### 4.5. Combination of Gene Expression and Copy Number Variation

The identification of differentially expressed genes with CNVs such as gain or loss was accomplished here, by considering the outputs of the gene expression analysis, so that, if the gene was upregulated or downregulated, and of CNV analysis, that is, if the region was amplified or deleted. We selected the upregulated genes with an amplified copy number and downregulated genes with copy number deletion in their corresponding regulatory enhancer regions of pancreatic adenocarcinoma patients. These genes correspond with the nature of the enhancer element.

### 4.6. Gene Network Construction, GO Enrichment and Super Enhancer Based Regulation

In order to understand the significance of both upregulated and downregulated genes, the STRING database [43] was used to construct the gene network of enhancer CNV driven genes at FDR 0.25. GO enrichment analysis of genes was done with the help of Enrichr, an R package [44]. We also aimed to identify super-enhancer-based gene regulation. Super-enhancers were defined as multiple enhancers regulating the same gene expression with an upregulating effect [45].

### 4.7. Correlation Analysis of Enhancer Expression with Patient Survival

Enhancer elements specifically expressed in pancreatic tissue were taken from study [3]. Pancreatic adenocarcinoma clinical files were downloaded from TCGA and correlation for enhancer expression and patient survival was done using in-house Python scripts.

## Figures and Tables

**Figure 1 ijms-20-03582-f001:**
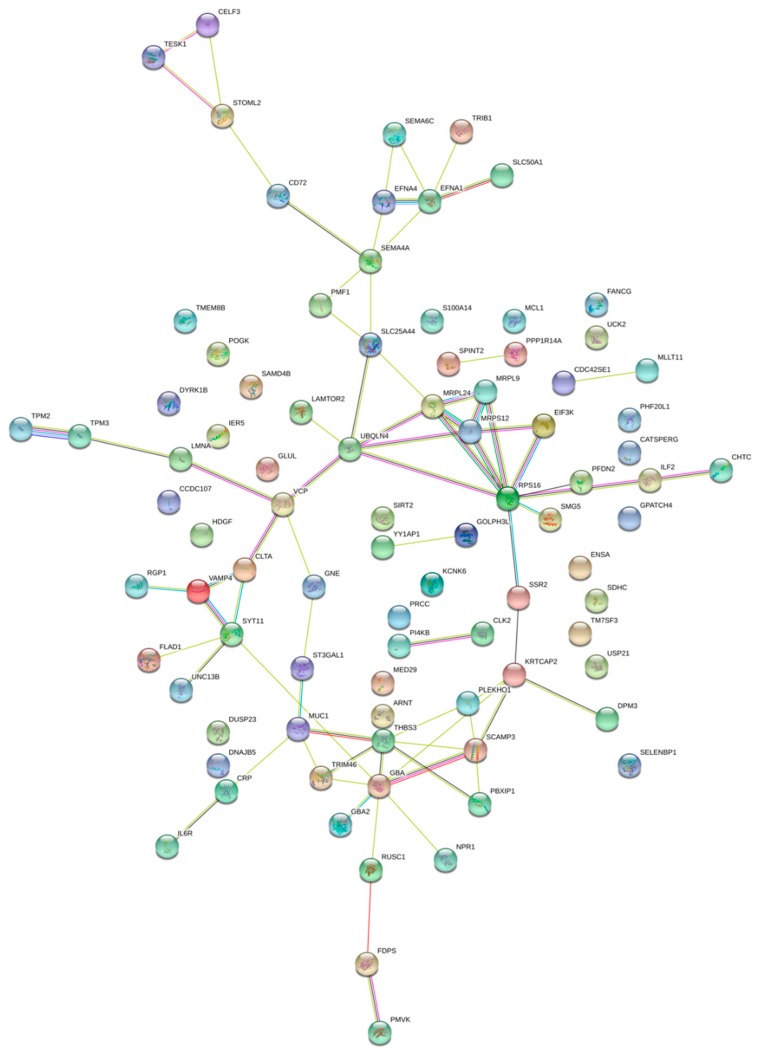
Gene interaction network of genes upregulated due to copy number gain in their regulatory enhancer region. Nodes represent proteins and edges represent protein–protein associations.

**Figure 2 ijms-20-03582-f002:**
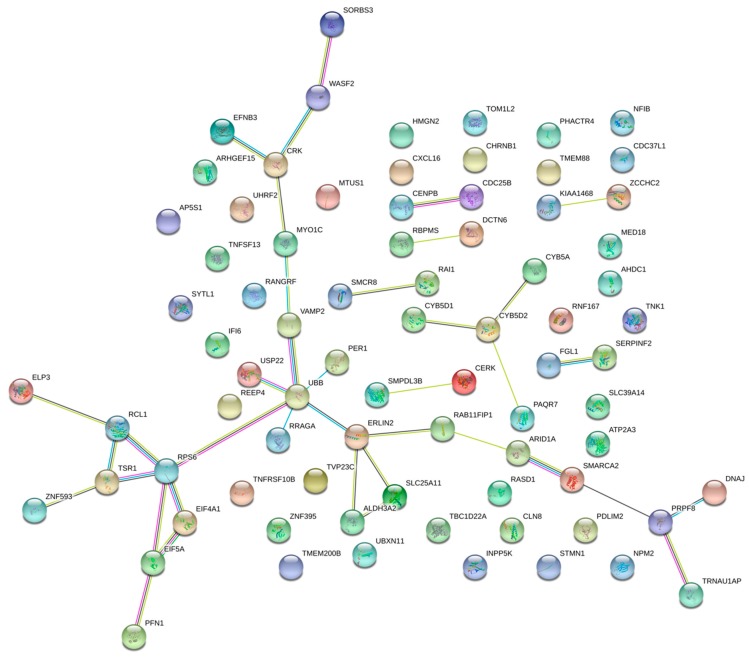
Gene interaction network of genes downregulated due to copy number loss in their regulatory enhancer region. Nodes represent proteins and edges represent protein–protein associations.

**Figure 3 ijms-20-03582-f003:**
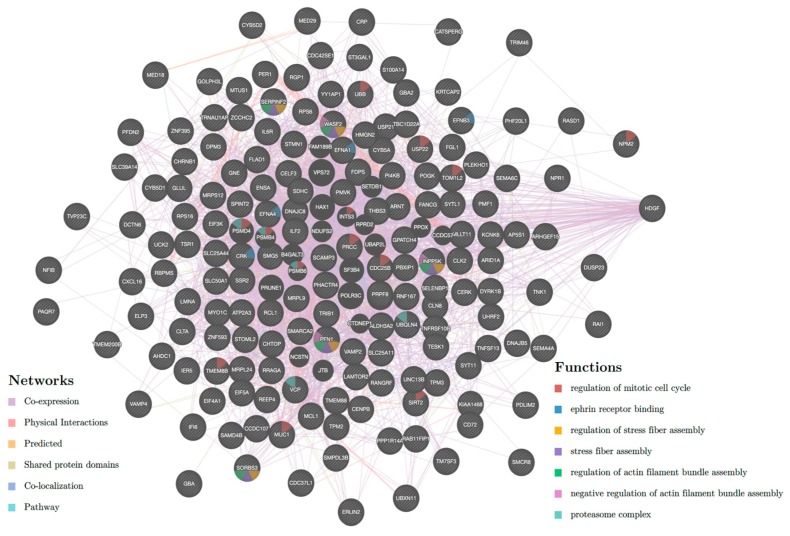
Pathway enrichment analysis performed using software Enrichr on 169 genes regulated by enhancers.

**Figure 4 ijms-20-03582-f004:**
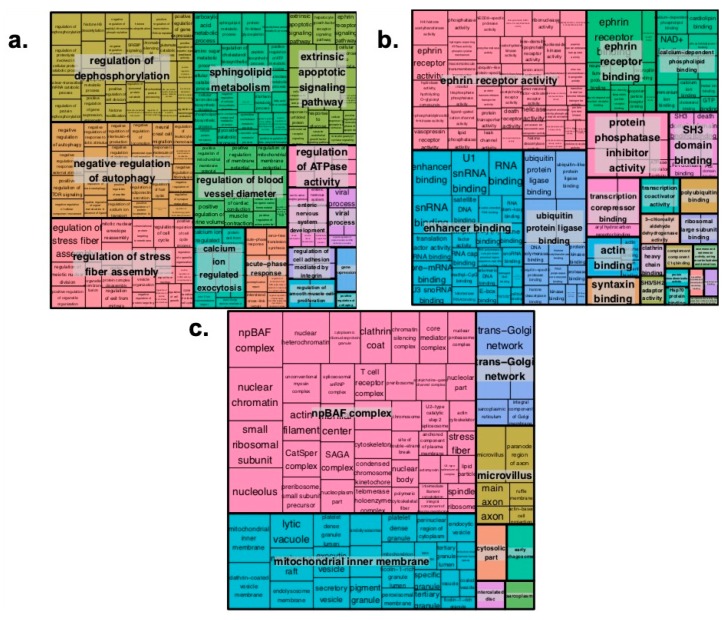
Tree map gene ontology analysis of enhancer CNV associated genes: (**a**) biological process; (**b**) molecular function; and, (**c**) cellular component.

**Figure 5 ijms-20-03582-f005:**
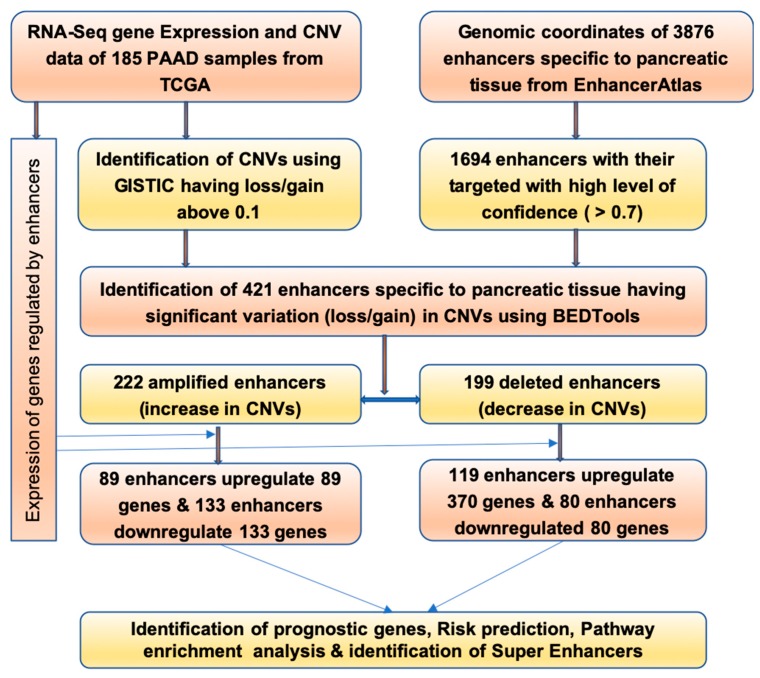
Computational workflow of the study.

**Table 1 ijms-20-03582-t001:** Cox proportion hazard ratios with *p*-values of the top 15 genes of enhancer regulated genes; genes sorted based on *p*-value.

Gene Symbol	Hazard Ratio (Median Value)	*p*-Value
*VAMP2*	2.917023	0.000221728
*GBA2*	2.641652475	0.000497427
*RANGRF*	2.512853346	0.000794802
*CERK*	2.483219821	0.001137485
*SORBS3*	2.450014903	0.001099829
*AP5S1*	2.375174975	0.001570686
*INPP5K*	2.317435264	0.00206786
*SLC25A44*	2.291295263	0.002326752
*RNF167*	2.239662092	0.003430129
*DPM3*	2.196709622	0.004076415
*SEMA6C*	2.192865982	0.005221015
*NPR1*	2.170189229	0.00497616
*SLC25A11*	2.104895785	0.006106619
*USP22*	2.093982578	0.006492927
*DNAJB5*	2.058641673	0.008559552

**Table 2 ijms-20-03582-t002:** Enhancer regulated genes present in the PRECOG database with z-score greater than 1.96. *Z*-scores of these genes were taken from the PRECOG database.

Gene	Name	*Z*-Score
*CLK2*	CDC-like kinase 2	2.39128
*EIF4A1*	Eukaryotic translation initiation factor 4A1	2.24365
*FDPS*	Farnesyl diphosphate synthase	4.03062
*FLAD1*	FAD1 flavin adenine dinucleotide synthetase homolog	2.95895
*HDGF*	Hepatoma-derived growth factor	2.16932
*ILF2*	Interleukin enhancer binding factor 2, 45kDa	2.80112
*MLLT11*	Myeloid/lymphoid or mixed-lineage leukemia (trithorax homolog, Drosophila); translocated to, 11	2.43407
*PFN1*	Profilin 1	2.49119
*PRCC*	Papillary renal cell carcinoma (translocation-associated)	3.93837
*REEP4*	Receptor accessory protein 4	2.98392
*RUSC1*	RUN and SH3 domain containing 1	3.28846
*SCAMP3*	Secretory carrier membrane protein 3	2.32083
*UBB*	In multiple clusters	2.54009
*UBQLN4*	Ubiquilin 4	2.79194
*USP21*	Ubiquitin specific peptidase 21	2.40871

**Table 3 ijms-20-03582-t003:** Genes regulated by super enhancers, including name of the gene, the genome coordinates of its enhancers and name of the chromosome.

Gene Ensembl ID	Enhancer Coordinates	Chromosome
ENSG00000010278	6337690-6341770;6387100-6388270;6449480-6450940;6579320-6579700	Chromosome 12
ENSG00000068745	49028850-49029210;49044960-49046260;49044960-49046260;49486410-49486940;49486410-49486940	Chromosome 3
ENSG00000099622	1248660-1249890;1248660-1249890;1248660-1249890;1250530-1251560;1259140-1263220;1275830-1277180;1275830-1277180;1275830-1277180;1275830-1277180	Chromosome 19
ENSG00000099875	1941420-1943940;2031500-2033130;2047440-2050790;2047440-2050790	Chromosome 19
ENSG00000111319	6444030-6449420;6449480-6450940;6449480-6450940;6662010-6662810	Chromosome 12
ENSG00000111674	6662010-6662810;6999750-7001150;6999750-7001150;7046720-7047010;7047230-7047600	Chromosome 12
ENSG00000114353	50126460-50126820;50328720-50329650;50358380-50358800;50359200-50359520;50359200-50359520	Chromosome 3
ENSG00000115524	198055790-198057570;198152000-198152560;198318390-198319050;198318390-198319050	Chromosome 2
ENSG00000116285	8181000-8181800;8181000-8181800;8193790-8194520;8318930-8320010	Chromosome 1
ENSG00000116473	112134680-112136260;112134680-112136260;112134680-112136260;112202750-112203810	Chromosome 1
ENSG00000117632	26221860-26223380;26221860-26223380;26323300-26324510;26452760-26454880	Chromosome 1
ENSG00000129968	1875340-1876950;1905440-1906150;1905440-1906150;2042030-2042430;2166660-2167720;2579200-2579650	Chromosome 19
ENSG00000130005	1040200-1040500;1040580-1040860;1383940-1385280;1407800-1410200	Chromosome 19
ENSG00000137154	19183280-19185190;19232860-19233120;19379620-19380190;19456550-19457860	Chromosome 9
ENSG00000142910	32013850-32015900;32109070-32109980;32109070-32109980;111308340-111308980;111948770-111949470	Chromosome 1
ENSG00000143294	156659170-156659710;156716040-156721140;156659170-156659710;156716040-156721140	Chromosome 1

**Table 4 ijms-20-03582-t004:** Correlation between CNV of enhancers and survival of patients in the case of cancer patients and healthy individuals. The red-to-green color gradient shows high to low correlation.

Coordinates of Enhancer on Genome	Correlation (CNV vs. Survival Time)	
	Cancer	Healthy
chr19:2059605-2060167	0.49837818	0.2314589
chr5:172380663-172381064	0.46715342	0.16699777
chr9:136999790-136999893	0.46557032	−0.0626969
chr16:85496879-85497321	0.45373278	−0.6866357
chr14:102415545-102415736	0.44115981	0.2792461
chr5:10352620-10352938	0.43075964	0.52245209
chr1:234746093-234747674	0.42760743	0.98845399
chr19:863983-864016	0.42492329	−0.3898893
chr10:31892273-31892723	0.42465511	−0.5260193
chr4:124621616-124621886	0.41837579	0.17175469
chr5:964244-964536	−0.1737289	0.52446177
chr6:169573934-169574190	−0.1747893	−0.7388221
chr22:50980817-50981280	−0.1776918	0.05853257
chr8:128306934-128307283	−0.1778833	−0.3632054
chr14:105500629-105500990	−0.1798565	0.11625874
chr15:99992993-99993428	−0.1823302	0.31824116
chrX:100792680-100793554	−0.1832455	−0.6606904
chr22:50979060-50979802	−0.1916855	0.01347895
chr18:12306995-12307375	−0.1946683	0.65842313
chr6:169569548-169569688	−0.1974295	0.1498992
chr8:70042378-70042779	−0.2421095	0.22802202

**Table 5 ijms-20-03582-t005:** Shows Cox proportion hazard ratio with *p*-value of top negatively correlated genes having potential of prognostic biomarkers, genes sorted based on *p*-value.

Negatively correlated genes (E^+^G^−^/E^−^G^+^)		
Gene Symbol	Hazard Ratio	*p*-Value
*MUM1*	3.558401305	1.33 × 10^−5^
*MUM1.1*	3.558401305	1.33 × 10^−5^
*MUM1.2*	3.558401305	1.33 × 10^−5^
*RBM6*	3.444153767	1.83 × 10^−5^
*MBD3*	3.027911665	0.00010983
*TLE2*	2.983689472	0.000127766
*TLE2.1*	2.983689472	0.000127766
*PLD3*	2.853819471	0.00022269
*PLD3.1*	2.853819471	0.00022269
*LRP3*	2.81420074	0.00030683

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
