# Peer review of "In Silico Analysis of Gene Expression Change Associated with Copy Number of Enhancers in Pancreatic Adenocarcinoma"

_ijms, 2019, doi:10.3390/ijms20143582_

Round 1

Reviewer 1 Report

1) Explain in more detail how enhancer and gene associations have been inferred and are defined
from the given set that was used in the study (He et al 2014).

2) Motivate in great detail in the introduction the strategy to
focus on amplified enhancer elements and corresponding genes which
are also amplified. Also explain the assumptions that are being made

by the selection of amplified enhancers and amplified genes? 

For example the gene and enhancer are located in the same amplified segment.

For example emphasize and cite related studies.
https://www.ncbi.nlm.nih.gov/pmc/articles/PMC4857881/
Zhang 2016, Nature Genet Identification of focally amplified lineage-specific super-enhancers in human epithelial cancers

6) Add missing details of the analysis that seem not to be described well

-- Number of samples/patients considered for the analysis
Tumor / Normal comparison

-- How were differential copy-number gain and loss defined for the analysis?
Describe in much greater detail. E.g. also emphazise
only patients with matched tumor normal sample pairs.
What percentage of patients have the selected amplifications that were selected?

3) Another major weakness of the study
is that results are presented in a single PAAD cohort that is available from TCGA. Other PAAD cohorts with clinical information are available and should be considered for the analysis
even if the concordance between the results is expected to be low, but it would tremendously increase the value of the study to consider validation of results.

For example identify any additional PAAD datasets with copynumber and/or gene expression profiles and or survival information
for validation of individual results (CNV, differential expression, functional analysis and survival).

For example microarray gene-expression profiles of 45 matching
pairs of pancreatic tumor and adjacent non-tumor tissues from 45 patients with pancreatic ductal adenocarcinoma
https://www.ncbi.nlm.nih.gov/geo/query/acc.cgi?acc=GSE28735
see also a relatively large collection in Table 1 in https://www.ncbi.nlm.nih.gov/pmc/articles/PMC5975421/

4) Survival analysis should be performed across multiple PAAD cohorts that are available.

5) Comparative analysis (e.g., survival, gene expression and other that are relevant)
required to show the difference between different
gene groups for the matched tumor/normal comparisons to show the properties
of the selected genes between the identified subset.

copy-number amplified genes CNV+,
non-copy number amplified genes CNV-
amplified enhancers EH+,
non-amplified enhancers EH-

This involves groups of CNV+/EH+, CNV-/EH+, etc.between differential expressed

and non-differential expressed genes.

The english language requires to be polished by e.g, a native english speaker.

Author Response

1) Explain in more detail how enhancer and gene associations have been inferred and are defined from the given set that was used in the study (He et al 2014).

Response 1: We are thankful to reviewer for useful suggestion. In revised version, we have modified Figure 5, to describe all steps in detail that include steps used to infers association between and gene expression. We also describe all steps in detail in Materials and Methods. We obtained 3876 enhances specific to pancreatic tissue from the EnhancerAtlas, a repository of all human enhancers that were identified through three or more independent high throughputs experimental evidence. In addition, EnhancerAtlas provided the target genes corresponding to these enhancers. A detailed of methodology of working enhancer-gene set used to perform this study is explained in the methodology section of the manuscript. In the revised version of manuscript, we have added the URL of the EnhancerAtlas used to download enhances and associated gene data. 

2) Motivate in great detail in the introduction the strategy to
focus on amplified enhancer elements and corresponding genes which
are also amplified. Also explain the assumptions that are being made by the selection of amplified enhancers and amplified genes? 

For example the gene and enhancer are located in the same amplified segment.

For example emphasize and cite related studies.
https://www.ncbi.nlm.nih.gov/pmc/articles/PMC4857881/
Zhang 2016, Nature Genet Identification of focally amplified lineage-specific super-enhancers in human epithelial cancers

Response 2:  We incorporate suggestion of reviewer in our revised manuscript. Copy number change is an important driver of tumorigenesis, this hypothesis is further supported by experimental evidences, such as increase in expression of MYC gene in case of lung cancer, increase in expression of KLF5 gene by amplification of corresponding regulatory element in case of squamous cell carcinoma. In the introduction section of the revised version of the manuscript, we have added the few evidences, which shows increase in gene expression when there is increase in copy number of their regulatory element.  

3) Another major weakness of the study is that results are presented in a single PAAD cohort that is available from TCGA. Other PAAD cohorts with clinical information are available and should be considered for the analysis even if the concordance between the results is expected to be low, but it would tremendously increase the value of the study to consider validation of results.

For example identify any additional PAAD datasets with copy number and/or gene expression profiles and or survival information for validation of individual results (CNV, differential expression, functional analysis and survival).

For example microarray gene-expression profiles of 45 matching
pairs of pancreatic tumor and adjacent non-tumor tissues from 45 patients with pancreatic ductal adenocarcinoma https://www.ncbi.nlm.nih.gov/geo/query/acc.cgi?acc=GSE28735
see also a relatively large collection in Table 1 in.
https://www.ncbi.nlm.nih.gov/pmc/articles/PMC5975421/

4) Survival analysis should be performed across multiple PAAD cohorts that are available.

Response 3 & 4: We fully agree with reviewer that this type of study should also be performed on  other PAAD cohorts. We searched different resources and databases to obtained PAAD cohorts, we only got one PAAD cohort used in this study. Unfortunately, we do not get another PAAD cohort data for performing analysis described in this manuscript.  GEO ID (GSE28735) including PubMed ID provided by reviewer only contain gene expression data which is in not sufficient to perform analysis described in our manuscript. The main focus of the present work is to understand effect of copy number change i.e. gain/loss in regulatory element on gene expressions of patients. Thus for our study, we need both gene expression and copy number variation in enhancers in cancer samples.  In revised manuscripts, we discussed suggestion of reviewer for future studies.

5) Comparative analysis (e.g., survival, gene expression and other that are relevant)
required to show the difference between different gene groups for the matched tumor/normal comparisons to show the properties of the selected genes between the identified subset.

copy-number amplified genes CNV+,
non-copy number amplified genes CNV-
amplified enhancers EH+,
non-amplified enhancers EH-

This involves groups of CNV+/EH+, CNV-/EH+, etc. between differential expressed and non-differential expressed genes.

Response to reviewer: We are thankful to reviewer for additional analysis.  As suggested by reviewer, create two sets and performed analysis on each set. Set 1 contain genes, which corresponds to the enhancer nature i.e. any change in enhancer region, effects the corresponding gene expression (E+G+/E-G-) and set 2 contains genes that do not corresponds to the enhancer nature they are negatively correlated with enhancer (E+G-/E-G+). The table of significant genes related to survival is provide in the revised manuscript. Only top ten result are shown in table as well as in manuscript, full table is provided in the supplementary file.

Gene   symbol

Hazard   Ratio

P-value

VAMP2

2.917023

0.00022173

GBA2

2.64165248

0.00049743

RANGRF

2.51285335

0.0007948

CERK

2.48321982

0.00113749

SORBS3

2.4500149

0.00109983

HDGF

0.51125711

0.01503922

EFNA4

0.50623421

0.01803272

SSR2

0.50020282

0.01113774

TM7SF3

0.46554002

0.00539798

DUSP23

0.44782961

0.00419565

Gene   symbol

Hazard   Ratio

P-value

MUM1

3.55840131

1.33E-05

MUM1.1

3.55840131

1.33E-05

MUM1.2

3.55840131

1.33E-05

RBM6

3.44415377

1.83E-05

MBD3

3.02791167

0.00010983

CSNK2A1

0.41286381

0.0011943

SEC24B

0.39980205

0.00086692

BZW1

0.37632248

0.00039711

BZW1.1

0.37632248

0.00039711

F3

0.33089441

7.39E-05

6) Add missing details of the analysis that seem not to be described well

-- Number of samples/patients considered for the analysis
Tumor / Normal comparison

Response: We are very thankful o the reviewer for this comment. The total number of sample used for this study are 185, Out of which only 4 samples are of normal as present in TCGA-PAAD.  

-- How were differential copy-number gain and loss defined for the analysis?

Response: The threshold used to define criteria the copy number loss or gain was 0.1 such as, regions with a copy number value above 0.1 acknowledged as copy number gain i.e. amplification, regions with a copy number value below 0.1 acknowledged copy number loss i.e. deletions.  

Describe in much greater detail. E.g. also emphasise  only patients with matched tumor normal sample pairs. What percentage of patients have the selected amplifications that were selected?

Response:  All the normal sample present in TCGA-PAAD are derived from adjacent solid tissue normal, having TCGA ID TCGA-H6-8124-11A-01R-2404-07, TCGA-H6-A45N-11A-12R-A26U-07, TCGA-HV-A5A3-11A-11R-A26U-07, TCGA-YB-A89D-11A-11R-A36G-07.

Additional comment of Reviewer 1

The english language requires to be polished by e.g, a native english speaker.

Response to reviewer: We are highly thankful to the reviewer for improving the quality of language of the manuscript. The revised manuscript is checked by Grammarly, for grammar related errors.

Reviewer 2 Report

The authors for this paper have done a comprehensive analysis of DNA-protein interactome (enhancer) for pancreatic cancer. The idea of investigating the relationship between gene expression change and copy number of enhancers is pretty straightforward. I have a few questions about this paper.

In section 2.5, where does the healthy sample comes from? What is the sample size for them?

Some figure legends are necessary for figure 1 and 2. What is the color for each node? Why do some edges have double lines?

Figure 3, many genes are blocked by those in the front.

Figure 4 is hard to read.

Author Response

Reviewer 2

The authors for this paper have done a comprehensive analysis of DNA-protein interactome (enhancer) for pancreatic cancer. The idea of investigating the relationship between gene expression change and copy number of enhancers is pretty straightforward. I have a few questions about this paper.

In section 2.5, where does the healthy sample comes from? What is the sample size for them?

Response: The healthy sample are present in the same TCGA-PAAD RNASeq data file. All the 4 normal sample present in TCGA-PAAD are derived from adjacent solid tissue normal, having TCGA ID TCGA-H6-8124-11A-01R-2404-07, TCGA-H6-A45N-11A-12R-A26U-07, TCGA-HV-A5A3-11A-11R-A26U-07, TCGA-YB-A89D-11A-11R-A36G-07. In the section 4.2 of revised version of the manuscript, we have added the normal sample information.

Some figure legends are necessary for figure 1 and 2. What is the color for each node? Why do some edges have double lines?

Response: We are thankful for the reviewer for providing valuable comment, so to improve the visibility of the figure legends. In the revised version of the manuscript, we have modified the figure legend of Figure 1 and Figure 2 and highlighted in yellow colour. Nodes represents the protein and color of each node represents interaction with other proteins. Edges in figure represents protein-protein associations confirmed by experimental and predicted studies. Double lines in edges represents that interaction between the proteins is confirmed by experimental as well as predicted studies. More information is provided in the string webserver, we are providing the link for the same (http://string-db.org/cgi/show_network_section.pl?taskId=C747sI2EeFsd&sessionId=bSjYsf8nWI0J&bottom_page_content=settings).

Figure 3, many genes are blocked by those in the front.

Response: In the revised manuscript, figures has been modified to improve the visibility.

Figure 4 is hard to read.

Response: In the revised manuscript, figures has been modified to improve the visibility.